# Plant and Disease Recognition Based on PMF Pipeline Domain Adaptation Method: Using Bark Images as Meta-Dataset

**DOI:** 10.3390/plants12183280

**Published:** 2023-09-15

**Authors:** Zhelin Cui, Kanglong Li, Chunyan Kang, Yi Wu, Tao Li, Mingyang Li

**Affiliations:** Co-Innovation Center for Sustainable Forestry in Southern China, Nanjing Forestry University, Nanjing 210037, China; 3210100096@njfu.edu.cn (Z.C.); likanglong@njfu.edu.cn (K.L.); kangchunyan@njfu.edu.cn (C.K.); wuyi051@njfu.edu.cn (Y.W.); litao3014@njfu.edu.cn (T.L.)

**Keywords:** image classification, few-shot learning, transfer learning, bark images dataset

## Abstract

Efficient image recognition is important in crop and forest management. However, it faces many challenges, such as the large number of plant species and diseases, the variability of plant appearance, and the scarcity of labeled data for training. To address this issue, we modified a SOTA Cross-Domain Few-shot Learning (CDFSL) method based on prototypical networks and attention mechanisms. We employed attention mechanisms to perform feature extraction and prototype generation by focusing on the most relevant parts of the images, then used prototypical networks to learn the prototype of each category and classify new instances. Finally, we demonstrated the effectiveness of the modified CDFSL method on several plant and disease recognition datasets. The results showed that the modified pipeline was able to recognize several cross-domain datasets using generic representations, and achieved up to 96.95% and 94.07% classification accuracy on datasets with the same and different domains, respectively. In addition, we visualized the experimental results, demonstrating the model’s stable transfer capability between datasets and the model’s high visual correlation with plant and disease biological characteristics. Moreover, by extending the classes of different semantics within the training dataset, our model can be generalized to other domains, which implies broad applicability.

## 1. Introduction

Image recognition technology based on artificial intelligence can provide scientific decision-making basis and optimization solutions by analyzing and processing images. This technology is of great importance to crop and forest management. However, its application faces many challenges, such as difficulties in data collection, the large number of classes, the variability of plant appearance, the difficulty of lesion detection, the invasion of new pathogens, and the impact of climate change [1].

Machine learning has been widely used in various agriculture and plant science domains [2], such as plant breeding [3], in vitro culture [4], stress phenotyping [5], stress physiology [6], plant system biology [7], plant identification [8], plant genetic engineering [9], and pathogen identification [10]. However, traditional machine learning methods have shortcomings in feature extraction, model selection, and data processing, which make it difficult to learn high-dimensional, non-linear, and unstructured data [11]. With the rapid development of computer science, deep learning began to appear. Deep learning refers to the use of deep neural networks to perform operations, such as automatic feature extraction and data classification, to achieve a high-level understanding and representation of features [12,13,14]. In agriculture and forestry, deep learning also provides effective technical methods to solve various computer visual tasks, such as plant pest and disease detection, forest inventory, plant classification and segmentation, and real-time monitoring of crop and forest resources, etc. [1,15,16,17,18,19,20].

While deep learning has been applied to image recognition in related domains, it often poses challenges when training neural networks for large-scale image recognition capabilities [21,22,23]. For instance, deep learning models often demand substantial training data, constraining their use in data-scarce or costly domains. They are also intricate, hindering transparency and validation of their internal mechanisms and learning strategies. Moreover, they can be sensitive to data distribution and noise, leading to errors in predicting new or abnormal images. To solve these problems, FSL (Few-shot Learning) has emerged [24,25,26,27,28]. FSL is a machine learning paradigm that aims to learn from a few labeled examples and generalize to new categories or domains [29]. FSL methods address these issues by leveraging prior knowledge and generalizing to novel categories, enabling effective image classification from limited or unlabeled data. FSL methods are suitable for plant disease recognition because they reduce data collection and labeling efforts and improve the model’s adaptability and robustness. 

The application of FSL in agriculture is mainly focused on plant and disease recognition, which is an important task for crop management and protection [1,30,31]. Most of the studies have been conducted by exploiting different feature extraction, data augmentation, metric learning, and self-supervised training strategies to improve the accuracy, robustness, and generalization of FSL models. These methods have been applied in both spatial and frequency domains, covering image classification and target detection tasks [31,32,33,34,35,36,37,38,39,40]. Chen et al. used a meta-learning framework for the adaptive process of the FSL plant disease detection task. Lin et al. proposed an FSL method for plant disease recognition based on multi-scale feature fusion and channel attention, and they also proposed different training strategies for different generalization scenarios. Pascual et al. analyzed the accuracy, robustness, and clustering ability of four FSL methods for fungal plant disease classification, with the prototypical network achieving the best results in accuracy and robustness. Lin et al. proposed an FSL algorithm for plant disease classification in the frequency domain, which outperforms existing methods in the spatial domain. Tunga et al. proposed a method to classify different corps, plants automatically, and their diseases based on the combination of the FSL algorithm and Transformers. Liu et al. used a self-supervised training strategy to generate pseudo-labels on a plant disease dataset and used the labels to fine-tune the prototypical network, achieving better results than supervised methods. Li et al. proposed an FSL framework for plant disease classification using limited labeled data that combines data augmentation, feature extraction, and metric learning. Zhang et al. proposed a data-driven FSL approach for crop disease and pest detection based on target detection and transfer learning. On the other hand, the applications of FSL in forestry are fewer, and most of the studies are based on remote sensing images for classification, such as hyperspectral image classification of tree species [41,42,43]. 

These studies demonstrate the effectiveness and the potential of FSL methods for plant and disease recognition. However, most previous research has focused on image recognition of species within a single domain. While these studies have contributed significantly to our understanding of specific agricultural or forestry applications, there is a noticeable gap in research that extends beyond these single domains. In real-world scenarios, plant and disease recognition often requires cross-domain adaptation, where the source domain (the labeled training set) and the target domain (the unlabeled test set) have different distributions. This poses a great challenge for traditional FSL methods, which may suffer from domain shift and over-fitting problems. Therefore, we plan to implement a Cross-Domain Few-shot Learning (CDFSL) image classification model for tree species classification and recognition of common plant and crop diseases (e.g., phytophthora, anthracnose, etc.). By extending our research across domains, we can obtain a more comprehensive understanding of the capabilities and limitations of FSL in these domains, ultimately providing agricultural and forestry operators with more functional and effective solutions.

PMF pipeline achieved state-of-the-art results on various CDFSL benchmarks, such as mini-ImageNet and Meta-Dataset [44]. In our study, we adapted and optimized the PMF pipeline to make it more suitable for plant and disease recognition. To test the performance of the tuned pipeline in different domains, we meta-trained and fine-tuned several models using BarkNetV3 and BarkVN50 datasets, respectively, and evaluated and visualized the effectiveness of the models on several datasets in the same and different domains. The research objectives include: (1) compare the performance of PMF of various frameworks on several novel datasets; (2) visualize the visual attractiveness of networks and analyze the inner workings of the mechanism; (3) analyze the generalization ability of PMF to the same and different domain datasets; (4) discuss the learning ability and practical application value of PMF for plant and disease recognition.

## 2. Materials and Methods

### 2.1. Datasets

The dataset for FSL is somewhat different from the common image classification task in deep learning. Traditional deep learning methods usually require a large amount of labeled data to train the model, while FSL aims to learn new categories from a few examples (usually no more than 10). Therefore, the datasets for FSL typically have the following characteristics: (1) the datasets contain multiple different data sources; (2) to simulate encountering a new classification situation in real work scenarios, there is no overlapping category in each subset; (3) the dataset provides test tasks of different difficulties. We show an example of an FSL image classification dataset, as shown in Figure 1.

In our study, we chose the tree bark image dataset as the meta-training dataset. We assume that bark images can be divided into low-level texture features and high-level color features, which serve as fine-grained data that can help the model learn more discriminative and pervasive representations for plant and disease recognition. That is, a more comprehensive recognition capability is obtained by learning complex datasets to predict relatively simple samples. We collected bark images of 20 different tree species on and around the campus of Nanjing Forestry University and named them BarkNJ. Our BarkNJ dataset has higher image quality than BarkNet 1.0 [45], which contains more noise, such as light, blur, or shadows. Noise affects feature extraction and label correlation, making it hard for the model to learn an efficient classifier from a few samples [22]. Nevertheless, we believe combining bark images of different qualities can enhance the overall generalization ability of the model in various scenarios. This is because field surveys of forests or crops have a complex environment that affects the quality of the image (e.g., lighting, angle, etc.). Therefore, we merged the high-quality BarkNJ dataset with BarkNet 1.0 into a more extensive dataset called BarkNetV3.

To test the meta-trained model, we introduced several publicly available image datasets in agriculture and forestry [46,47,48,49,50] to explore the performance of the FSL model on datasets from the same domain or different domains. We processed these datasets into meta-datasets with the following steps: (1) Remove images with high similarity or ambiguous features. (2) Classes with less than 40 images are removed, and classes with too many images (>600) have the dataset processed by (1), and then 600 images are randomly selected to be used as research data. The image dataset used in our study is shown in Table 1.

### 2.2. Methodology

In the PMF pipeline, “PMF” stands for three different training phases, where “P”, “M”, and “F” are taken from the first letter of Pre-train, Meta-train, and Fine-tune, respectively. Pre-train finds useful features in external data by self-supervised learning of image representation. Meta-train uses ProtoNet (Prototypical Network) to acquire representations of images and map correlations between categories in a high-dimensional space, thus creating a general method for recognizing data in the same or different domains. Fine-tune, on the other hand, accomplishes learning and transfer from novel datasets by fine-tuning the model and augmenting the support set. Instead of developing a new FSL algorithm, we adapt and optimize its structure based on the PMF pipeline for use in our research area, as shown in Figure 2.

#### 2.2.1. Backbone of Pre-Train

Backbone refers to the feature extraction network, which is used as the basis for various tasks such as semantic segmentation, instance segmentation, and object detection [12,24,53]. By leveraging pre-trained model weights trained on large datasets like ImageNet as encoders, we exploit the wealth of diverse image information in ImageNet. This initialization strategy is critical to mitigate overfitting and enhance the model’s ability to effectively generalize across classes with few examples, thereby supporting the success of ProtoNet in few-shot image classification.

Four network pre-training weights are used in this paper to compare the performance of different backbones in our experiments, all of which are trained from ImageNet-1k. DINO-ViT and DeiT-ViT are self-supervised training approaches employing Vision Transformers (ViT) [54] for learning robust and transferable visual representations from large-scale unlabeled data. DINO [55] emphasizes DIstillation with NO labels, while DeiT [56] is short for Data-efficient image Transformers. ResNet (Residual Network) and its variants [57,58] are a common backbone for FSL classification. DINO-ResNet is a self-supervised method that learns visual features from unlabeled images without contrastive learning or distillation.

#### 2.2.2. Algorithm of Meta-Train

ProtoNet (Prototypical Network) is an FSL method that performs classification by learning an embedding space and then measuring distances to obtain a prototypical representation of each class [59]. The ProtoNet consists of two parts: an embedding function and a distance function [60]. We defined a process for a single train step as follows: (1) We used an encoder network that mapped the input data to a latent space (such as feature vectors). (2) We adopted a prototype layer [61,62] that computed the mean vector of each class in the latent space, serving as the prototype representation of that class. (3) We used cosine similarity to measure the similarity between a query point and the prototypes. (4) We introduced pseudo-labels for each episode to simulate the data distribution and scarcity in the FSL classification scenario [63,64], enhancing the quality and interpretability of the low-dimensional embedding by incorporating prior information from a semi-supervised learning algorithm. (5) In addition, we introduced the prototype refining method proposed by [65,66]. Prototype refining was the process of updating the prototypes using both labeled and unlabeled instances in each iteration of transductive inference.

#### 2.2.3. Task-Specific Fine-Tune

Task-specific Fine-tuning is the process of adapting and updating the weights of a pre-trained feature extractor to fit a specific task [28,30,44]. The FSL model achieves new sample classification by using a few labeled examples (support set) from the new task and allows the meta-trained network to learn the features of the new task [60,66]. The process of fine-tuning is as follows: (1) extraction of pertinent features from the support set and query set using the pre-trained neural network; (2) derivation of class prototypes, which are representative examples for each category in the support set; (3) computation of similarity scores between query features and prototypes; and (4) assignment of logits, indicating the likelihood of query examples belonging to each support set category. Since a small amount of data may cause instability in model training, we also explored the method of feature aggregation [67,68,69] and data augmentation [70] for multiple support sets of each category separately to create more robust category prototypes. In addition, we apply feature regularization techniques to reduce over-fitting.

Task-specific Fine-tuning can improve the performance of FSL algorithms, which is especially effective when the input data is in a different domain from the source data [71]. Through fine-tuning, models can alleviate the instability caused by domain shifts. By using a small number of images in the support set, the fine-tuned model can generalize the recognition features of a specific new task, thereby making predictions on data in different domains.

#### 2.2.4. Visualization

(1)T-distributed stochastic neighbor embedding (t-SNE). t-SNE is a technique for visualizing high-dimensional data in a low-dimensional space [72]. It aims to preserve the local structure of the data by minimizing the Kullback-Leibler divergence between the joint probabilities of the high-dimensional data and the low-dimensional embedding. By using pseudo-labels as conditional variables, t-SNE can discount the known structure of the data given the true labels and highlight the conditional structure of the data given the pseudo-labels [63,64]. This can reveal more meaningful and relevant patterns in the embedding, such as clusters, outliers, or subgroups that the standard t-SNE may not capture. Pseudo-labels can also help to reduce the noise and ambiguity in the high-dimensional data and improve the stability and robustness of the t-SNE optimization.(2)Smooth Grad CAM++. It is an advanced variation of the popular Grad-CAM (Gradient-weighted Class Activation Mapping) technique [73]. It is used in deep learning and computer vision to visualize and interpret the decision-making process of neural networks by highlighting regions in an input image that influence the network’s classification or prediction. Smooth Grad CAM++ can also visualize different layers, feature maps, or neurons within a model at the inference level, which means when the model is making predictions.

### 2.3. Parameters Setting

Different types of datasets also require additional considerations for CDFSL. For example, we use two types of datasets in our experiments: multi-classification datasets with different semantics and those with the same semantics. The former kind of dataset, such as mini-ImageNet, contains images of various objects from different semantics, such as animals, plants, flowers, and architectures. The latter kind of dataset contains images of objects from the same category, such as our bark datasets. This kind of dataset, where the classes are related in meaning and appearance, poses a challenge for assessing the ability of FSL models to distinguish between similar tasks that are hard to differentiate. Moreover, since the bark dataset’s composition is simpler than the mini-ImageNet, our training process may face a more severe over-fitting problem. Thus, we need to reconsider the performance of the backbone and the prototypical network.

The meta-training parameters were adjusted as shown in Table 2, which includes (1) shortening the total number of training epochs; (2) adjusting the change rule of the learning rate for each epoch according to different backbone and datasets, and due to the low total number of epochs trained, we appropriately increased the ratio of decay-epochs, warmup-epochs, cooldown-epochs, and patience-epochs; (3) significantly reducing the number of episodes per epoch to prevent over-fitting or performance decay; and (4) to maintain the stability of the training process, turning off the precision mixing and random erasure, and reducing the magnitude of data augmentation; (5) since the features in our dataset are relatively simple, we tend to choose smaller backbone in our experiments, and the selected backbones are DINO-ViT-small, DeiT-ViT-small, ResNet50, and DINO-ResNet50.

## 3. Results and Analysis

### 3.1. Meta-Train

In our experiments, DINO-ViT shows the best performance when meta-training on both bark datasets. As shown in Table 3, DINO-ViT achieves an accuracy of 82.81% and 74.08% on BarkVN50 and BarkNetV3, respectively. The classification accuracy of DeiT-ViT is second only to DINO-ViT, achieving 80.37% and 72.26% on the two bark datasets, respectively. The meta-training classification accuracy of the two ViT-based self-supervised classification models we used is better than those with traditional ResNet as the backbone by about six percent on average. The classification accuracy of DINO-ResNet is located in the middle between DeiT-ViT and ResNet, by about three percent higher.

By comparing the training results of the four backbone models on two bark datasets (Table 3), the results show that (1) DINO-ViT achieves the best performance on both datasets and both scenarios, indicating that vision transformers are superior to convolutional or hybrid networks for few-shot image classification, as they can learn more generalizable features from unlabeled data using self-supervised learning; (2) BarkVN50 is easier than BarkNetV3 for few-shot image classification, as all models obtain higher accuracy and lower loss on BarkVN50 than on BarkNetV3. This may be because BarkVN50 was collected from Southeast Asian rainforests, which means more diversity and less similarity among categories; (3) the saturation point varies among models and datasets, ranging from the 20th to 30th epoch. This means that fine-tuning strategies and convergence metrics should be adapted to account for these variations, optimize performance, and ensure efficient use of computational resources. 

### 3.2. Meta-Test

We tested the models obtained in our experiments on six datasets from the same and different domains to assess how well each model generalizes in novel datasets and to investigate whether certain models tend to overfit known categories, leading to degraded performance in new categories (Table 4). 

Accuracy and loss are two metrics that measure how well a machine learning model predicts the correct output. In terms of classification accuracy, the model trained on the BarkVN50 dataset (referred to as the BarkVN50 model, so it is with other models) performs the best in most test datasets, both in 1-shot and 5-shot. The mini-ImageNet and the BarkNetV3 model perform slightly lower than the BarkVN50 model, but the difference is slight (about 2–3%). This implies that the models have similar performance when they are tested on a different but related domain, which shares some common-use features and classes with the source domain. That is expected due to the similarity of data distribution and label space. The Full-Dataset model performs worst on out-of-domain datasets, except for the Flowers dataset. This implies that simply combining different source domains into one dataset does not improve the performance of CDFSL and may even degrade it due to over-fitting or conflicting information.

In terms of loss, the BarkVN50 model is well-fitted to the BarkVN50 dataset, but it may not generalize well to other datasets, especially those that are out-of-domain or have different types of images. The mini-ImageNet and Full-Dataset models, on the other hand, although less effective in predicting bark images, slightly outperform the BarkVN50 and BarkNetV3 models on some agriculture datasets. It is worth mentioning that the BarkNetV3 model achieves a lower loss than the other models on most of the test datasets. The BarkNetV3 dataset, which this model is trained on, may have more diverse and complex features than the other datasets, enabling the BarkNetV3 model to learn more generalizable and robust features that can transfer well to other plant and disease recognition datasets, even those that are out-of-domain or have different types of images. This may account for the lower losses of the BarkNetV3 model on most datasets.

### 3.3. Visualization

To simulate the output of each test set, we present the results of our model as a pseudo-classification map, as illustrated in Figure 3, which depicts the model’s learning and transfer capability on novel datasets. The performance of models in handling new tasks is closely associated with the discreteness observed within the distribution of pseudo-classes. In other words, when the distribution exhibits greater separation and distinctiveness among pseudo-classes, the model’s adaptability to novel tasks is markedly enhanced. 

The results showed that the BarkVN50 model achieved good classification results on most datasets, with mostly significant segmentation between pseudo-classes, except on the Agricultural Disease dataset, where it performed poorly. On the other hand, the BarkNetV3 model showed its reliability by producing robust pseudo-classification maps on different forestry datasets, both in-domain and out-of-domain. The pseudo-classification maps of this model showed a high degree of discreteness, indicating that the BarkNetV3 model was more stable and adaptable in transfer learning than the BarkVN50 model. Since the dataset used for pre-training may have lacked some common-use features of agriculture and forestry images, the performance of the mini-ImageNet and Full-Dataset models was slightly worse than the other two models, with a weaker ability to discriminate between pseudo-classes. This suggested that the applicability of these two models in plant and disease recognition may have been somewhat lower. It was worth mentioning that both mini-ImageNet and Full-Dataset models had good processing capability for cross-domain datasets like PlantVillage and Flowers. Therefore, these models may have been more suitable for some FSL classification tasks with more obvious distinguishing features (compared to bark). To further illustrate the effectiveness of our trained model, we visualized the recognition process of the network using Smooth Grad CAM++, as shown in Figure 4. We found that visual attractiveness was highly correlated with the locations of phenotypic plant diseases that occurred biologically, indicating that the model tended to focus more on regions where the plant displayed symptoms of diseases, such as spots, lesions, discoloration, or deformation.

In the training process of FSL image classification models, the pursuit of enhanced accuracy often leads to various training strategies. However, these strategies may not always improve the performance of the models and may even cause over-training and degradation. To systematically evaluate the consequences of over-training on FSL learning models, we conducted a deliberate experiment by extending the training duration to 30 and 50 epochs, corresponding to episode numbers of 200 and 500, respectively. While this extended training procedure yielded a modest increase in accuracy of about 2.5 percent in our experiments, it also revealed a critical trade-off. Despite the numerical improvement in accuracy, the predictive ability of the model trained in this way showed a significant decline, as visually depicted in Figure 5. Figure 5 provides a visual representation of the performance degradation observed in the BarkNetV3 and BarkVN50 models due to over-training. Notably, these t-SNE plots reveal significant confusion between pseudo-classes, reflecting a compromised ability of the model to discriminate between classes effectively. Furthermore, the loss values during over-training significantly increase, on average, about 20 percent higher than those of standard training methods. This phenomenon highlights the importance of careful model training strategies to avoid over-fitting, calls for a refined approach to balancing accuracy and generalization in FSL classification tasks, and demonstrates the effectiveness of our parameters setting.

## 4. Discussion

In this paper, we demonstrated that our meta-trained model could recognize unseen tree species and achieved high accuracy on various plant and disease datasets after fine-tuning. The results showed that the modified PMF pipeline was able to recognize several cross-domain datasets using generic representations and demonstrated high visual relevance. Moreover, by extending the classes of different semantics within the training dataset, our model can also be generalized to other domains, which implies broad applicability.

Our experiments were meta-trained using four backbones, in which DINO-ViT had the highest training accuracy. We speculate that it is based on the following reasons: (1) The structure of DINO-ViT enables it to adapt to new domains and categories with only a small number of labeled examples, eliminating the need for extensive retraining or domain adaptation. (2) Unlike traditional self-supervised learning methods that require a large memory bank to store negative samples, DINO-ViT uses no contrastive loss or dictionary. This reduces the dependence on large-scale labeled data, which is impractical for FSL scenarios. (3) DINO-ViT captures global context and long-range dependencies more effectively than traditional convolutional neural networks such as ResNet. The performance of DeiT-ViT is slightly lower than DINO-ViT. This may be because DeiT-ViT typically requires a large amount of labeled data and a robust CNN teacher network to achieve good results. Thus, the backbone of using DeiT-ViT for FSL may face the problem of attentional collapse, where the model focuses on only a few tokens and ignores the others, thus hindering its ability to capture the global context. In addition, although DINO-ResNet combines DINO and ResNet, DINO-ResNet is still inferior to DINO-ViT in most downstream tasks. This may be because the model is constrained by the shortcomings of ResNet, such as limited receptive fields, spatial resolution, etc., and thus is less capable than DINO-ViT in terms of transfer ability. Notably, instead of optimizing parameters for training accuracy, our experiments emphasized the model’s ability to generalize to both the test set and new datasets within the same and different domains.

Figure 3 and Table 4 show the test results of the four FSL models through quantification and visualization, respectively. Since the dataset may lack some potential common-use features of crop images, the BarkVN50 model can not process Agricultural Diseases efficiently, and the generated images are significantly more chaotic than other types of datasets. Similarly, the mini-ImageNet and Full-Dataest models have poor classification capabilities for bark images due to the lack of some common-use features of tree bark images. However, the test result of these two models is relatively good for recognizing flowers and crop diseases. This may be because these datasets contain some images of flowers and leaves, so the trained model has a specific classification ability for such images. It is worth mentioning that the BarkNetV3 model achieves more consistent results both for predictions in the same domain (bark) and different domains (plants and diseases), with clear segmentation lines between the pseudo-classes. This indicates that the BarkNetV3 model may be more suitable for plant and disease recognition. Therefore, although mini-ImageNet is considered a more generalized dataset, it may not be as good as specialized datasets for classification in some specific domains. In particular, none of the four models tested well against BarkKR. This may be due to the small size of this dataset and the prevalent existence of images that contain some noise, such as buildings, roads, sidewalks, and cars, which causes the network to fail to capture the key representations. 

In addition, we analyzed the visual attractiveness of the network using Smooth Grad CAM++, and most lesion occurrence locations were highly correlated with hotspots. This could be because these regions have more distinctive features that can help the model discriminate between different classes of diseases. Alternatively, this could be because these regions have more salient features that can attract the model’s attention. In either case, this finding suggests that the model has learned some useful information about the occurrence and distribution of plant diseases from the training data, and is also suitable for practical field-based disease recognition applications. We believe that the success of the model is due to the fact that bark images have rich texture and color features that can help the model learn more discriminative and generic representations of plants. Tree bark textures can be regarded as foundational, capturing intricate, fine-grained details: (1) texture features can capture the fine-grained details and patterns of the bark surface, such as cracks, scales, or ridges, which can be useful phenotypic features for identifying different plants or diseases; (2) color features can reflect the variations and changes in the bark color, such as green, brown, or yellow, indicating different plant health conditions or disease symptoms. By fine-tuning the models to transfer these foundational features, they become adept at recognizing higher-level attributes associated with crops, plants, and diseases. This hierarchical integration of features makes it easier for the models to understand the interplay between the bark textures and the agricultural elements.

Compared to previous studies [30,31,32,33,34,35,36,37,38,39,40], our experiments achieved similar or even better results in terms of (1) the accuracy of prediction, despite being trained from bark, our tests on some public datasets (such as Agricultural Disease, PlantVillage, and Flowers) yielded promising results, with an average 5-shot accuracy of about 93%; (2) the ability of domain adaptation; while other methods may rely on more specific or domain-dependent features, our method can adapt to different regions, environments, and seasons more effectively than other methods; (3) the amount of data required (e.g., BarkVN50 has only 4000 images), reduce the cost and time for data collection and annotation; and (4) the transfer capability, as shown in the t-SNE visualization, the performance of the model is more stable in the transfer between domains. It is important to note that, unlike previous studies on FSL in agriculture, our work focuses on CDFSL. However, there are fewer studies in this area, so the comparisons with similar work may not be comprehensive. 

Nevertheless, the application value of CDFSL image classification in plant and disease recognition is considerably broad. When staff need to recognize a species that is not included in the dataset, they only need to input five labeled images as learning samples, and the fine-tuned model can be applied to the recognition of this new category. Using this technique, staff can collect images in the field and later upload them to the server for recognition. Through optimizing data acquisition and image processing, our FSL model can meet the needs of fieldwork in terms of efficiency and accuracy. In addition, by accumulating a large amount of data and model optimization, the model’s generalization performance can be continuously improved, and the model can be transferred to applications in different scenarios.

Plant and disease recognition covers a wide range of species and symptoms, and our experiments are insufficient to generalize the features of all the classes. Thus, our method still has some drawbacks: (1) we did not test on a CDFSL image dataset covering multiple semantics classes in other domains, which limits the ability to use our trained models in specific domains; (2) our method may not be able to handle some complex or rare plants and diseases that require more specialized knowledge or features; (3) our method may not be able to capture some contextual or temporal information that may affect plant health or disease diagnosis; (4) the predictions or reasoning process in a transparent or interpretable way. In our future research, we will utilize prior knowledge from similar domains as auxiliary data to enhance both the data efficiency and generalization capability of our model, while also conducting further optimizations of the FSL algorithm.

## 5. Conclusions

In this paper, we modified an effective pipeline of CDFSL for plant and disease recognition, and analyzed and visualized the model’s performance on multiple datasets from the same and different domains. In our experiments, we exploited the rich texture and color features of bark images to learn more discriminative and generalizable representations for plant and disease recognition. In addition, we used some visualization techniques to analyze the stability of the model on the novel dataset, as well as the recognition process of the neural network. We evaluated our method on various plant and disease datasets and obtained similar or even better results than previous studies in terms of the ability of domain adaptation, the amount of data required, and the transfer capability. Furthermore, we demonstrated the effectiveness of the recognition process using Grad CAM, revealing a strong correlation between the feature location of plant diseases and visual attractiveness. Based on our modified PMF pipeline, integrating diverse images with various agricultural and forestry semantics into the meta-dataset can enhance the model’s ability to generalize comprehensive recognition features, expanding its applicability to a broader range of application scenarios.

## Figures and Tables

**Figure 1 plants-12-03280-f001:**
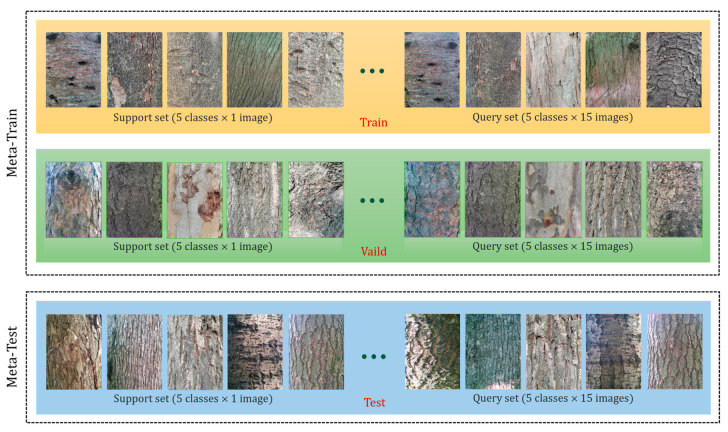
An example of an FSL dataset (5-way 1-shot). To evaluate the performance of the FSL model, a common way is N-way and K-shot, i.e., there are N categories to be classified, and each category has K-labeled images as the support set. The model needs to use the support set to learn the characteristics and similarities of the categories and then classify new images of the same class in the query set. There are usually 15 images in each category in the query set, and the model needs to correctly predict the labels of these images based on the support set. A support set and a query set are called an episode, and the model’s accuracy is computed by averaging the results of multiple combinations (a total of episodes) of support and query sets randomly selected from the dataset.

**Figure 2 plants-12-03280-f002:**
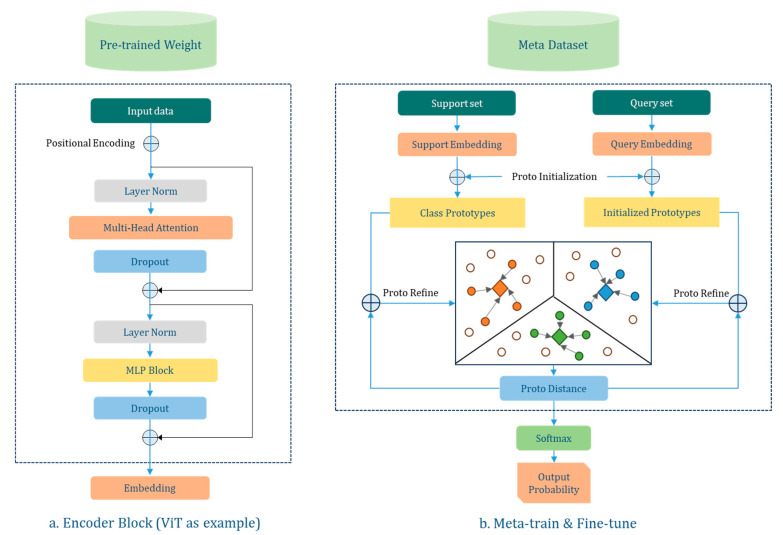
Schematic diagram of PMF pipeline. (**a**) shows the Vision Transformers (ViT) encoder, and (**b**) shows the meta-training and fine-tuning process using ProtoNet as the FSL algorithm. It should be noted that (1) The output of (**a**) will be used as support embedding and query embedding in (**b**); (2) we use data from the support set to construct pseudo-labels through data augmentation and compute prototypes of categories based on the pseudo-labels; and (3) in addition, for each new task, we only update the backbone network parameters instead of exploring the applicability of the model.

**Figure 3 plants-12-03280-f003:**
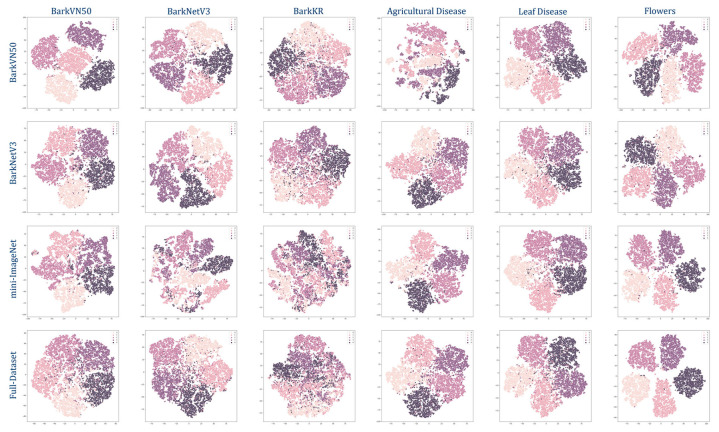
Task-specific Fine-tune pseudo-labeling classification chart (5-shot). The caption on the left indicates the dataset used for training, and the caption above the image indicates the dataset used for testing, both using the backbone DINO-ViT.

**Figure 4 plants-12-03280-f004:**
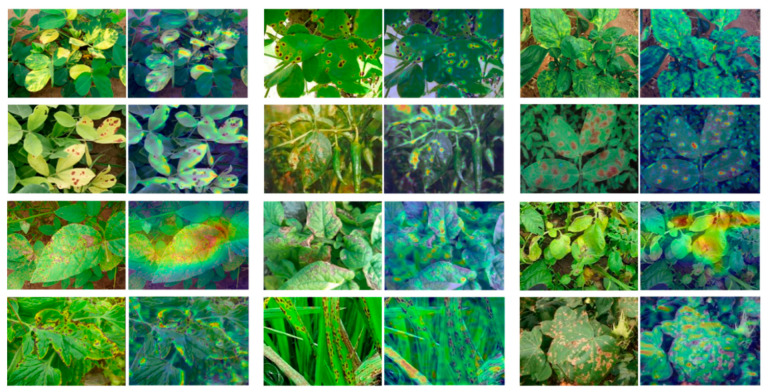
Class activation mapping generated via Smooth Grad CAM++. The image to the right of each input image is calculated by Smooth Grad CAM++, with the lesion portion highlighted by a heat map. These images were taken from publicly available disease images from the Chinese Academy of Agricultural Research and its affiliates. Moreover, we have specially selected some images with more complex backgrounds to confirm the effectiveness of our visual recognition process.

**Figure 5 plants-12-03280-f005:**
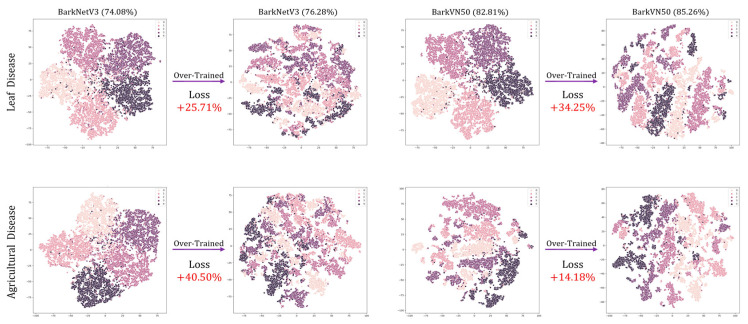
An example of over-training. The dataset and corresponding classification accuracy are labeled in “alpha (beta%)” format at the top of the images. Loss indicates the percentage increase in the loss value of the model after over-training. The top and bottom columns of the image show the changes in the test results of the models before and after over-trained on the PlantVillage and Agricultural Disease datasets, respectively.

**Table 1 plants-12-03280-t001:** Details of the dataset used in our experiment.

Domain	Dataset	Collaborators	Categories	Images	Meta-Dataset
Tree species classification	BarkNet 1.0	Carpentier et al. (2018) [45]	23	23,616	20 × 600
	BarkNJ	Ours (2023) [47]	20	14,681	20 × 600
	BarkNetV3	Ours (2023) [45,47]	40	24,000	40 × 600
	BarkVN50	Truong Hoang (2017) [51]	50	5578	50 × 80
	BarkKR	Tae Kyung et al. (2022) [46]	54	6918	25 × 50
Leaf Diseases	PlantVillage	Hughes et al. (2015) [48]	40	61,484	38 × 600
Crop Diseases	Agricultural Diseases	Xulang Guan et al. (2021) [50]	60	36,258	55 × 100
Flower Identification	Flowers 102	Nilsback et al. (2008) [49]	102	8189	85 × 40
Multi-Classification	mini-ImageNet	Vinyals et al. (2016) [52]	100	60,000	100 × 600
	Full-Dataset	Hu et al. (2022) [44]	8 datasets	-	-

Note: Domain denotes the specific domain of the dataset. Dataset is the name of the dataset. Collaborators are the collectors of the dataset. Categories and Images indicate the number of categories and total images in the dataset. Meta-dataset denotes the processed meta-dataset, and the “alpha*beta” represents the number of categories (alpha) and the number of images per category (beta) in the meta-dataset. In addition, mini-ImageNet [52] is a benchmark dataset for FSL image classification, and Full-Dataset [44] is a CDFSL dataset constructed by Hu et al.

**Table 2 plants-12-03280-t002:** Meta-training parameter settings.

Dataset	Backbone	nEpisode	Lr	DWCP (%)	nQuery	nClsEpisode
BarkVN50	DINO-ViT	(200, 300)	5 × 10^−4^	(20, 10, 10, 20)	15	5
	DeiT-ViT	(200, 300)	5 × 10^−4^	(20, 10, 10, 20)	15	5
	ResNet	(300, 500)	1 × 10^−4^	(20, 10, 20, 10)	15	5
	DINO-ResNet	(200, 300)	1 × 10^−4^	(20, 10, 20, 10)	15	5
BarkNetV3	DINO-ViT	(200, 500)	5 × 10^−4^	(10, 10, 10, 20)	15	5
	DeiT-ViT	(200, 500)	5 × 10^−4^	(10, 10, 10, 20)	15	5
	ResNet	(300, 500)	1 × 10^−4^	(10, 10, 20, 10)	15	5
	DINO-ResNet	(200, 500)	1 × 10^−4^	(10, 10, 20, 10)	15	5

Note: Dataset denotes the dataset used for training, Backbone denotes the pre-trained backbone (based on ImageNet) used, nEpisode denotes the number of episodes, Lr denotes the learning rate, DWCP denotes the proportion of training phases, nQuery is the number of images contained in each query set, nClsEpisode is the number of categories in each episode. Where nEpisode interval denotes the range of values for the number of episodes, with an interval of 100; DWCP is taken from the initial letters of Decay-epochs, Warmup-epochs, Cooldown-epochs, and Patience-epochs in turn, and the value in the DWCP (%) parentheses indicates the percentage of the total epoch that each phase represents.

**Table 3 plants-12-03280-t003:** Meta-training results.

Dataset	Backbone	Accuracy (1-Shot)	Loss (1-Shot)	Accuracy (5-Shot)	Loss (5-Shot)	Saturation
BarkVN50	DINO-ViT	71.81%	0.744	82.81%	0.504	25–26th
	DeiT-ViT	67.37%	0.893	80.37%	0.683	21–25th
	ResNet	60.59%	1.044	74.59%	0.754	24–28th
	DINO-ResNet	64.80%	1.139	77.80%	0.859	20–27th
BarkNetV3	DINO-ViT	62.08%	1.034	74.08%	0.774	26–28th
	DeiT-ViT	60.26%	1.103	72.26%	0.803	22–28th
	ResNet	57.78%	1.112	67.78%	0.862	24–30th
	DINO-ResNet	60.35%	1.217	71.35%	0.977	23–30th

Note: Accuracy and Loss denote accuracy and loss, respectively, on the validation set (not the test set), and 1-shot and 5-shot in parentheses represent the metrics evaluated (the number of images of each category extracted from each episode of the support set), respectively. Saturation indicates the point at which training peaks or saturates (when there is little drop in loss).

**Table 4 plants-12-03280-t004:** Task-specifically test results.

Train on	Test on	1-Shot (%)	Loss	5-Shot (%)	Loss
BarkVN50	BarkVN50 (I)	88.78 ± 0.99	0.339	96.95 ± 0.42	0.119
	BarkNetV3 (I)	81.64 ± 1.33	0.527	93.30 ± 0.73	0.242
	BarkKR (I)	63.28 ± 1.44	1.010	82.91 ± 1.07	0.494
	Agricultural Disease (O)	82.05 ± 1.43	0.508	94.07 ± 0.82	0.460
	PlantVillage (O)	81.57 ± 1.26	0.705	93.96 ± 0.65	0.332
	Flowers (O)	81.01 ± 1.34	0.592	95.37 ± 0.63	0.501
BarkNetV3	BarkVN50 (I)	77.24 ± 1.30	0.655	92.04 ± 0.68	0.319
	BarkNetV3 (I)	83.91 ± 1.17	0.449	93.39 ± 0.67	0.204
	BarkKR (I)	59.53 ± 1.54	1.136	80.21 ± 1.08	0.554
	Agricultural Disease (O)	76.21 ± 1.46	0.676	90.55 ± 0.95	0.354
	PlantVillage (O)	75.35 ± 1.41	0.778	90.15 ± 0.87	0.340
	Flowers (O)	78.47 ± 1.61	0.722	94.65 ± 0.61	0.467
mini-ImageNet	BarkVN50 (O)	79.73 ± 1.27	0.594	94.85 ± 0.55	0.447
	BarkNetV3 (O)	77.25 ± 1.49	0.707	91.47 ± 0.71	0.463
	BarkKR (O)	56.97 ± 1.50	0.567	79.70 ± 1.19	0.708
	Agricultural Disease (O)	79.95 ± 1.50	0.645	93.88 ± 0.86	0.318
	PlantVillage (O)	80.10 ± 1.34	0.753	92.33 ± 0.79	0.291
	Flowers (I)	83.84 ± 1.42	0.873	96.95 ± 0.39	0.252
Full-Dataset	BarkVN50 (O)	68.91 ± 1.35	0.864	88.79 ± 0.83	0.375
	BarkNetV3 (O)	62.88 ± 1.41	1.038	84.91 ± 0.96	0.468
	BarkKR (O)	51.86 ± 1.58	1.298	74.71 ± 1.32	0.712
	Agricultural Disease (O)	73.55 ± 1.63	0.724	91.01 ± 0.94	0.322
	PlantVillage (O)	68.57 ± 1.53	0.873	91.17 ± 0.88	0.346
	Flowers (I)	95.93 ± 0.64	0.152	98.97 ± 0.25	0.044

Note: The data in this table was obtained by sampling in the test set of each dataset 1000 times. “I” and “O” in parentheses of the dataset denote in domain and out domain, respectively; 1-shot and 5-shot indicate the number of images of each class passed to the network for each episode, e.g., 1-shot means that the network is only shown one image of that class and the prediction of the same class is made; the interval values below 1-shot and 5-shot denote the accuracy deviation on the test set after fine-tuning of the FSL network.

## Data Availability

Data available on request—The data underlying this article will be shared on reasonable request with the corresponding author.

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
