# Peer review of "Plant and Disease Recognition Based on PMF Pipeline Domain Adaptation Method: Using Bark Images as Meta-Dataset"

_plants, 2023, doi:10.3390/plants12183280_

Round 1
Reviewer 1 Report
The manuscript titled "Plant and Disease Recognition Based on PMF-Pipeline Domain Adaptation Method: Using Bark Images as Meta-Dataset" presents a study on using Few-Shot Learning (FSL) techniques for efficient image recognition in the context of plant and disease recognition. While the study is promising and addresses an important area of research, there are several key concerns that need to be addressed through a major revision before the manuscript can be considered for publication. The following comments outline the major issues that need to be addressed:
The abstract and content of the manuscript are somewhat convoluted and lack clarity. The manuscript needs to be restructured to clearly present the research objectives.
The introduction should clearly highlight the significance of the problem, the existing challenges in plant and disease recognition, and how Few-Shot Learning (FSL) methods aim to address these challenges. Additionally, it would be beneficial to introduce machine learning and different applications of ML in different fields of agriculture and plant science (lines 30-32). So, I suggest the following sentences:
Machine learning has been widely used in different fields of agriculture and plant science such as plant breeding (DOI: 10.1016/j.isci.2020.101890), in vitro culture (10.1007/s00253-020-10888-2), stress phenotyping (10.1016/j.tplants.2015.10.015), stress physiology (10.1371/journal.pone.0240427), plant system biology (10.1007/s00253-022-11963-6), plant identification (10.1016/j.compag.2016.07.003), plant genetic engineering (10.1371/journal.pone.0239901) and pathogen identification (https://doi.org/10.1094/MPMI-08-18-0221-FI).
Methodology: The manuscript briefly mentions the "PMF-Pipeline Domain Adaptation Method," but lacks sufficient details on its components and how it is applied. The methodology section should provide a comprehensive overview of the approach, including a step-by-step explanation of the PMF pipeline, how the FSL models were trained and fine-tuned, and the rationale behind using the BarkNetV3 and BarkVN50 datasets. Also, the authors missed figure 1. Please insert Figure 1.
Experimental Setup and Results: The experimental setup and results need to be better presented and organized.
The discussion should go beyond the numerical results and provide insights into the practical implications of the findings. Address how the proposed approach compares to existing methods, its limitations, and potential areas for improvement.
The manuscript contains several grammatical errors and awkward sentence structures that need to be addressed. Additionally, ensure that the technical terminology is accurate and consistent throughout the manuscript.
In conclusion, while the manuscript addresses an important research area, it requires a major revision to address the concerns outlined above. Improvements in clarity, organization, methodology presentation, experimental setup, and language usage are crucial for making this manuscript suitable for publication.
The manuscript contains several grammatical errors and awkward sentence structures that need to be addressed. Additionally, ensure that the technical terminology is accurate and consistent throughout the manuscript.
Author Response
Dear Editor and Reviewer,
Thank you very much for having our paper reviewed and sending us your comments, which were highly insightful and enabled us to improve the quality of our manuscript.

Reviewer 2 Report
Plant and Disease Recognition Based on PMF-Pipeline Domain Adaptation Method: Using Bark Images as Meta-Dataset is presented in this work. The following are the observations or comments that must be incorporated to strengthen the work.
There have been many works done in the past on plant disease prediction. So it is necessary to highlight the key contributions of this work and the novelty involved in that should be highlighted in the introduction part.
If the dataset is not collected by the authors, then its year should be mentioned.
Without much discussion about the proposed model and the workflow of the proposed model, this work largely focuses on result analysis which becomes meaningless.
Detailed algorithm description is required for the proposed model
Is the proposed model novel or an already used pre-trained model?
No other well-known models are considered for the comparison of results.
Technically, significant improvements are required to strengthen this work which lacks largely on the underlying research.
Author Response

(The authors gave the same response as above.)

Round 2
Reviewer 1 Report
All the comments have been addressed. I think that the current version of the manuscript can be published in Plants.
Reviewer 2 Report
The authors have addressed the comments and this work can be accepted